# Severe pulmonary complications after cytoreductive surgery and hyperthermic intraperitoneal chemotherapy are common and contribute to decreased overall survival

Olivia Sand[1], Mikael Andersson[2], Erebouni Arakelian[1‡], Peter Cashin[1‡], Egidijus Semenas[1‡], Wilhelm Graf[1]

1 Department of Surgical Sciences, Uppsala University Hospital, Uppsala, Sweden, 2 Department of Medical Sciences, Respiratory, Allergy and Sleep Research, Uppsala University, Uppsala, Sweden

☯ These authors contributed equally to this work.
‡ These authors also contributed equally to this work.
* olivia.sand@surgsci.uu.se

## Abstract

### Background and objectives

Extensive abdominal surgery is associated with the risk of postoperative pulmonary complications. This study aims to explore the incidence and risk factors for developing postoperative pulmonary complications after cytoreductive surgery and hyperthermic intraperitoneal chemotherapy and to analyze how these complications affect overall survival.

### Methods

Data were collected on 417 patients undergoing surgery between 2007 and 2017 at Uppsala University Hospital, Sweden. Postoperative pulmonary complications were graded according to the Clavien-Dindo classification system where Grade $\geq$ 3 was considered a severe complication. A logistic regression analysis was used to analyze risk factors for postoperative pulmonary complications and a Cox proportional hazards model to assess impact on survival.

### Results

Seventy-two patients (17%) developed severe postoperative pulmonary complications. Risk factors were full thickness diaphragmatic injury and/or diaphragmatic resection [OR 5.393, 95% CI 2.924–9.948, p = < 0.001]. Severe postoperative pulmonary complications, in combination with non-pulmonary complications, contributed to decreased overall survival [HR 2.285, 95% CI 1.232–4.241, p = 0.009].

### Conclusions

Severe postoperative pulmonary complications were common and contributed to decreased overall survival. Full thickness diaphragmatic injury and/or diaphragmatic resection were the

**Data Availability Statement:** The data underlying the results presented in this study contain potentially sensitive and identifying participant information and cannot be shared publicly due to GDPR. The data are available upon request from the Uppsala University registrar (registrator@uu. se; reference UU-DsO 2021/106).

**Funding:** The study was funded by the Swedish Cancer Society Grant Number 160411 and WG was the one receiving the research grant. https://www. cancerfonden.se/. The funders had no role in study design, data collection and analysis, decision to publish, or preparation of the manuscript.

**Competing interests:** The authors have declared that no competing interests exist.

**Abbreviations:** ASA score, American Society of Anesthesiologists' physical status classification; BMI, Body mass index; CC score, Completeness of cytoreduction score; CPAP, Continuous positive airway pressure; CRS, Cytoreductive surgery; HIPEC, Hyperthermic intraperitoneal chemotherapy; PCI, Peritoneal cancer index; PM, Peritoneal metastases; PPCs, Postoperative pulmonary complications.

main risk factors. This finding emphasizes the need for further research on the mechanisms behind pulmonary complications and their association with mortality.

## Introduction

Peritoneal metastasis (PM) represents the third most frequent metastatic site after hepatic and pulmonary deposits in colorectal cancer [1]. Historically, the diagnosis has been associated with a poor prognosis [2] but the 1990s saw the introduction of a curative treatment for selected patients with PM: cytoreductive surgery (CRS) and hyperthermic intraperitoneal chemotherapy (HIPEC) [3]. The aim of this procedure is to resect all visible tumor tissue and perioperatively flush the abdominal cavity with heated chemotherapy, targeting microscopic residual tumor tissue [3]. CRS and HIPEC has led to an increased survival rate in patients with PM [4] but postoperative morbidity and mortality rates still range between 10% and 30% and 1% and 5% respectively [5–8].

Postoperative pulmonary complications (PPCs), such as pneumonia and respiratory failure, are common in the period after abdominal surgery [9] and studies on PPCs after CRS and HIPEC quote an incidence between 6.8% and 69% [10–13]. Non-pulmonary complications such as hemorrhage and wound infection are also frequent after CRS and HIPEC: reports suggest that the overall severe morbidity rate is around 12% [8]. Some of the proposed risk factors for developing PPCs after abdominal surgery are poor general health status, smoking, lung disease, and the type and duration of surgery [9,14]. Suggested prophylactic measures include pre- and postoperative interventions such as smoking cessation, prophylactic physical therapy, and continuous positive airway pressure (CPAP) in the postoperative period [15–17]. A better understanding of the risk factors for developing PPCs after CRS and HIPEC could improve the perioperative treatment of patients with PM and potentially decrease the incidence of PPCs.

There is currently limited information regarding risk factors related to PPCs after CRS and HIPEC, and to what extent PPCs impact survival rates. The aim of this study was to determine the incidence and risk factors for developing severe PPCs after CRS and HIPEC. Another aim was to investigate whether PPCs impact overall survival.

## Materials and methods

### Patient selection and data collection

This was a cross-sectional study on all patients who underwent CRS and HIPEC at the Uppsala University Hospital between 2007 and 2017. Data were extracted from both medical records and the Swedish Intensive Care Unit Registry (PasIva) [18]. Demographic and clinical data were collected on gender, age, body mass index (BMI), lung disease (defined as chronic obstructive pulmonary disease, asthma, chronic bronchitis and/or sleep apnea), smoking (current or former smoker), the American Society of Anesthesiologists' physical status classification (ASA), and primary tumor origin. Data were also collected on operative variables such as the peritoneal cancer index (PCI) [19], completeness of cytoreduction score (CC score) [20], duration of surgery, liver resection, splenectomy, diaphragmatic peritonectomy, full thickness diaphragmatic injury and/or diaphragmatic resection, type and grade of pulmonary complications, as well as type and grade of non-pulmonary complications. The study was approved by the Swedish Ethical Review Authority in Uppsala, Sweden (reference no. 2013/203). The data

were analyzed anonymously and therefore, written/verbal consent was not required or obtained.

Patient selection for CRS and HIPEC is based on the absence of unresectable metastatic disease, PCI and patient ability to withstand extensive surgery [21]. To identify patients for inclusion, we used a local register from Uppsala University Hospital of patients who had undergone surgery because of PM. All 438 adult patients who underwent CRS and HIPEC between January 1, 2007 and December 31, 2017 were eligible for the study and the primary tumor origin was colon, rectum, appendix, small intestine, stomach, ovarium or peritoneum. The most frequently used chemotherapy regimens were Mitomycin C (for 90 minutes) for appendiceal tumors or Oxaliplatin (for 30 minutes) for tumors of colorectal origin.

## Grading of complications

In this study the Clavien-Dindo classification of surgical complications [22] was used to grade both pulmonary and non-pulmonary complications. The classification system consists of seven severity grades including two subgrades for Grades 3 and 4. Grade 1 refers to a deviation from the normal postoperative course without the need for pharmacological (with some exceptions), surgical, radiological or endoscopic treatment. Grade 2 corresponds to complications that need pharmacological treatment, blood transfusion or total parenteral nutrition. Complications that require surgical, endoscopic or radiological interventions are classified as Grade 3 and the two subcategories within Grade 3 divide the interventions into (a) interventions not under general anesthesia or (b) interventions under general anesthesia. Examples of a Grade 3 complication are pleural effusion that requires drainage and reoperation because of anastomotic leaks. A Grade 4 category corresponds to life-threatening complications requiring ICU management where single organ dysfunction is graded 4a and multi-organ dysfunction is graded 4b. In this study respiratory complications that required prolonged mechanical ventilation (defined as longer than what was routine) and/or reintubation or non-invasive ventilation due to respiratory complications were graded 4. Surgical complications that required intensive care were sepsis and various heart conditions. Death is graded 5. In this study, we classified PPCs Grade $\geq$ 3 as being severe PPCs. For the purpose of this study, patients were divided into two groups based on the occurrence of PPCs: patients with PPCs Grade < 3 (including those with no PPCs) and patients with PPCs Grade $\geq$ 3 according to the Clavien-Dindo classification of surgical complications. All PPCs occurring during the hospital stay at Uppsala University Hospital were identified and graded. All non-pulmonary complications were grouped together and split into two groups: Grade < 3 or Grade $\geq$ 3.

## Statistical analysis

Demographic, clinical and operative characteristics were presented descriptively in relation to the presence or absence of severe PPCs. Continuous data were presented with means and standard deviations. Categorical data were presented with frequencies and percentages. A univariate analysis of risk factors considered to be associated with severe PPCs was performed. Variables with a p-value $\leq$ 0.1 in the univariate analysis and variables previously well established as risk factors were included in the multivariate logistic regression model and the odds ratio (OR) for each variable was estimated. A Kaplan-Meier estimation including a log rank test and a Cox proportional hazards model was performed to analyze survival rates between patients with different grades of PPCs and/or non-pulmonary complications. Variable selection was performed by conducting a univariate analysis of variables considered to be associated with overall survival. A Cox proportional hazards model was then built from the variables with a p-value $\leq$ 0.1 in the univariate analysis and variables known to be correlated with overall

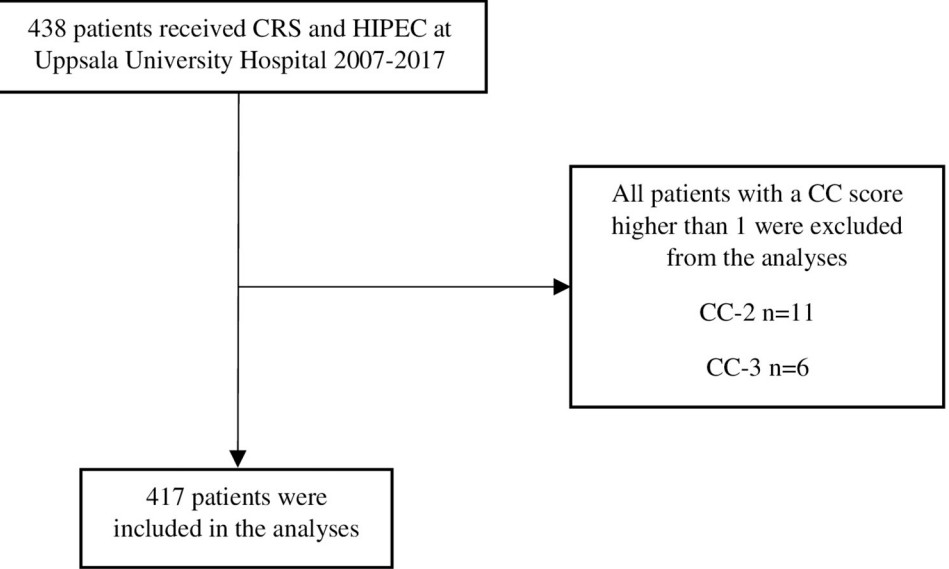

**Fig 1. Flowchart of inclusions and exclusions of the study.** *CRS* Cytoreductive surgery *HIPEC* Hyperthermic intraperitoneal chemotherapy *CC score* Completeness of cytoreduction score.

survival. Overall survival was defined as the time from surgery to date of death. Patients who were alive at the end of the data collection (April 25, 2019) were censored. All data were processed and analyzed with IBM SPSS Statistics for Windows version 23 (IBM Corp, Armonk, NY, USA). The significance threshold was set at < 0.05.

## Results

### Baseline characteristics

A total of 438 patients underwent CRS and HIPEC between January 1, 2007 and December 31, 2017. After excluding patients with an unknown CC score (n = 4), CC-2 (n = 6) and CC-3 (n = 11), 417 patients were included in the analyses (Fig 1). Patient characteristics are presented in Table 1.

### Incidence of complications and risk factors for severe PPCs

The incidence of PPCs Grade $\geq$ 3 was 17% (72 out of 417 patients) and the types of PPCs were pleural effusion or pneumothorax requiring drainage (n = 60) and postoperative reintubation or prolonged need of mechanical ventilation/non-invasive ventilation because of pulmonary complications such as respiratory failure and pneumothorax (n = 25). The percentage of patients developing one or more non-pulmonary complications Grade $\geq$ 3 was 9% (38 out of 417 patients) and reoperation was required in 25 (66%) of these patients. The types of non-pulmonary complications Grade $\geq$ 3 were abdominal abscess (n = 15), hemorrhage (n = 11), cardiac complications (n = 6), gastrointestinal perforations (n = 6), sepsis (n = 5), wound dehiscence (n = 5), anastomotic leaks (n = 4) and ileus (n = 2). In all, 52/72 of PPCs Grade $\geq$ 3 were isolated pulmonary complications whereas 20 (28%) occurred together with non-pulmonary complications Grade $\geq$ 3 (Table 1). Full thickness diaphragmatic injury and/or diaphragmatic resection [OR 5.393, 95% CI 2.924–9.948, p = < 0.001] were the most influential variables in the multivariate model, while neither age, smoking, PCI, liver resection nor diaphragmatic peritonectomy contributed statistically significantly to the model (Table 2).

**Table 1. Characteristics of 417 patients undergoing CRS and HIPEC between 2007 and 2017.**

| | PPCs Grade <3 (n = 345) | PPCs Grade ≥3 (n = 72) |
|---|---|---|
| Gender (male) | 143 (41%) | 31 (43%) |
| Age, years | | |
| Mean, SD | 57 ± 13.0 | 60 ± 11.5 |
| BMI | 25.8 ± 4.1 | 26.6 ± 5.4 |
| ASA score | | |
| 1 | 95 (27.5%) | 14 (19.5%) |
| 2 | 192 (56%) | 42 (58%) |
| 3 | 45 (13%) | 13 (18%) |
| 4 | 5 (1.5%) | 2 (3%) |
| Missing | 8 (2%/) | 1 (1.5%) |
| Smoking[a] | 87 (25%) | 24 (33%) |
| Lung disease[b] | 26 (7.5%) | 7 (10%) |
| Primary tumor | | |
| Colon | 133 (38.5%) | 27 (37.5%) |
| Appendix | 119 (34.5%) | 21 (29%) |
| Rectum | 13 (4%) | 5 (7%) |
| Small intestine | 9 (3%) | 4 (5.5%) |
| Ovarium | 12 (3.5%) | 4 (5.5%) |
| Appendix (benign) | 7 (2%) | 0 (0%) |
| Gastric | 2 (0.5%) | 1 (1.5%) |
| Mesothelioma | 14 (4%) | 2 (3%) |
| Unknown primary | 36 (10%) | 8 (11%) |
| Diagnosis | | |
| PMP, no neoplastic cells | 153 (44%) | 25 (35%) |
| Colorectal, mesothelioma and others | 192 (56%) | 47 (65%) |
| PCI | 15.2 ± 10.5 | 20.0 ± 10.0 |
| Duration of surgery (min) | 470 ± 149.4 | 530 ± 137.3 |
| Liver resection | 65 (19%) | 23 (32%) |
| Splenectomy | 110 (32%) | 30 (42%) |
| Diaphragmatic peritonectomy | 171 (50%) | 55 (76%) |
| Full thickness diaphragmatic injury and/or diaphragmatic resection | 35 (10%) | 30 (42%) |
| CC score | | |
| 0 | 275 (80%) | 47 (65%) |
| 1 | 70 (20%) | 25 (35%) |
| Non-pulmonary complications[c] | | |
| No | 327 (95%) | 52 (72%) |
| Yes | 18 (5%) | 20 (28%) |

*CRS* Cytoreductive surgery *HIPEC* Hyperthermic intraperitoneal chemotherapy *PPCs* Postoperative pulmonary complications *SD* Standard deviation *BMI* Body mass index *ASA* American Society of Anesthesiologists.

*PCI* Peritoneal Cancer Index *CC score* Completeness of cytoreduction score.

[a] Current or former smoker.

[b] Chronic obstructive pulmonary disease, asthma, chronic bronchitis and/or sleep apnea.

[c] Including abdominal abscess, hemorrhage, cardiac complications, gastrointestinal perforations, sepsis, wound dehiscence, anastomotic leaks and ileus.

**Table 2. Unadjusted and adjusted results of odds ratios and p-values for the variables included in the final model on risk factors for severe pulmonary complications after CRS and HIPEC.**

| | Unadjusted results | P-value[a] | Adjusted results | P-value[a] |
|---|---|---|---|---|
| | OR (95% CI) | | OR (95% CI) | |
| Age | 1.000 (1.000–1.000) | 0.104 | 1.000 (1.000–1.000) | 0.095 |
| BMI | | | | |
| < 18.5 | 1.926 (0.369–10.045) | 0.437 | | |
| 18.5–25 | Reference | | | |
| > 25–30 | 1.219 (0.681–2.182) | 0.506 | | |
| > 30 | 1.712 (0.857–3.418) | 0.128 | | |
| ASA score | | | | |
| 1 | Reference | | | |
| 2 | 0.368 (0.065–2.085) | 0.259 | | |
| 3 | 0.547 (0.103–2.915) | 0.480 | | |
| 4 | 0.722 (0.125–4.165) | 0.716 | | |
| Smoking[b] | 1.483 (0.858–2.562) | 0.158 | 1.605 (0.866–2.975) | 0.133 |
| PCI | 1.043 (1.018–1.068) | 0.001 | 1.020 (0.986–1.054) | 0.255 |
| Liver resection | 2.022 (1.150–3.554) | 0.014 | 1.576 (0.831–2.991) | 0.164 |
| Splenectomy | 1.526 (0.907–2.568) | 0.111 | | |
| Diaphragmatic peritonectomy | 3.292 (1.837–5.900) | < 0.001 | 1.828 (0.872–3.832) | 0.110 |
| Full thickness diaphragmatic injury and/or diaphragmatic resection | 6.327 (3.526–11.351) | < 0.001 | 5.393 (2.924–9.948) | < 0.001 |

*CRS* Cytoreductive surgery *HIPEC* Hyperthermic surgery *OR* Odds ratio *CI* Confidence interval *BMI* Body mass index *ASA score* American Association of Anesthesiologists score *PCI* Peritoneal cancer index.

[a] Logistic regression.

[b] Current or former smoker.

## PPCs and overall survival

In the Cox proportional hazards model, PPCs Grade ≥3 were not independently associated with decreased overall survival (p = 0.3). However, having both PPCs Grade ≥3 and non-pulmonary complications Grade ≥3 was associated with decreased overall survival (Fig 2), [HR 2.285, 95% CI 1.232–4.241, p = 0.009] when adjusted for gender, age, BMI, ASA score, diagnosis, CC score and PCI (Table 3).

## Discussion

In this large cohort of patients, we found severe PPCs to be common after CRS and HIPEC and to contribute to decreased overall survival. The multivariate model of risk factors identified full thickness diaphragmatic injury and/or diaphragmatic resection as significantly associated with severe PPCs.

To date, our study is the largest one conducted on the incidence and risk factors for PPCs after CRS and HIPEC. Previous studies have presented a wide range of incidence numbers, 6.8%-69%, and have also used a variety of grading systems [10–13], which complicates comparison. Sinukumar et al. [10], Arakalian et al. [11], and Cascales Campos et al. [12] all used different versions of the Common Terminology Criteria for Adverse Events (CTCAE) [23] whereas Preti et al. [13] used a customized classification system for their study. We found a higher incidence of PPCs than the majority of these studies and we believe that this could be partially explained by the use of a different grading system. Unfortunately, we could find no previous studies on PPCs in patients with peritoneal metastases using the same classification

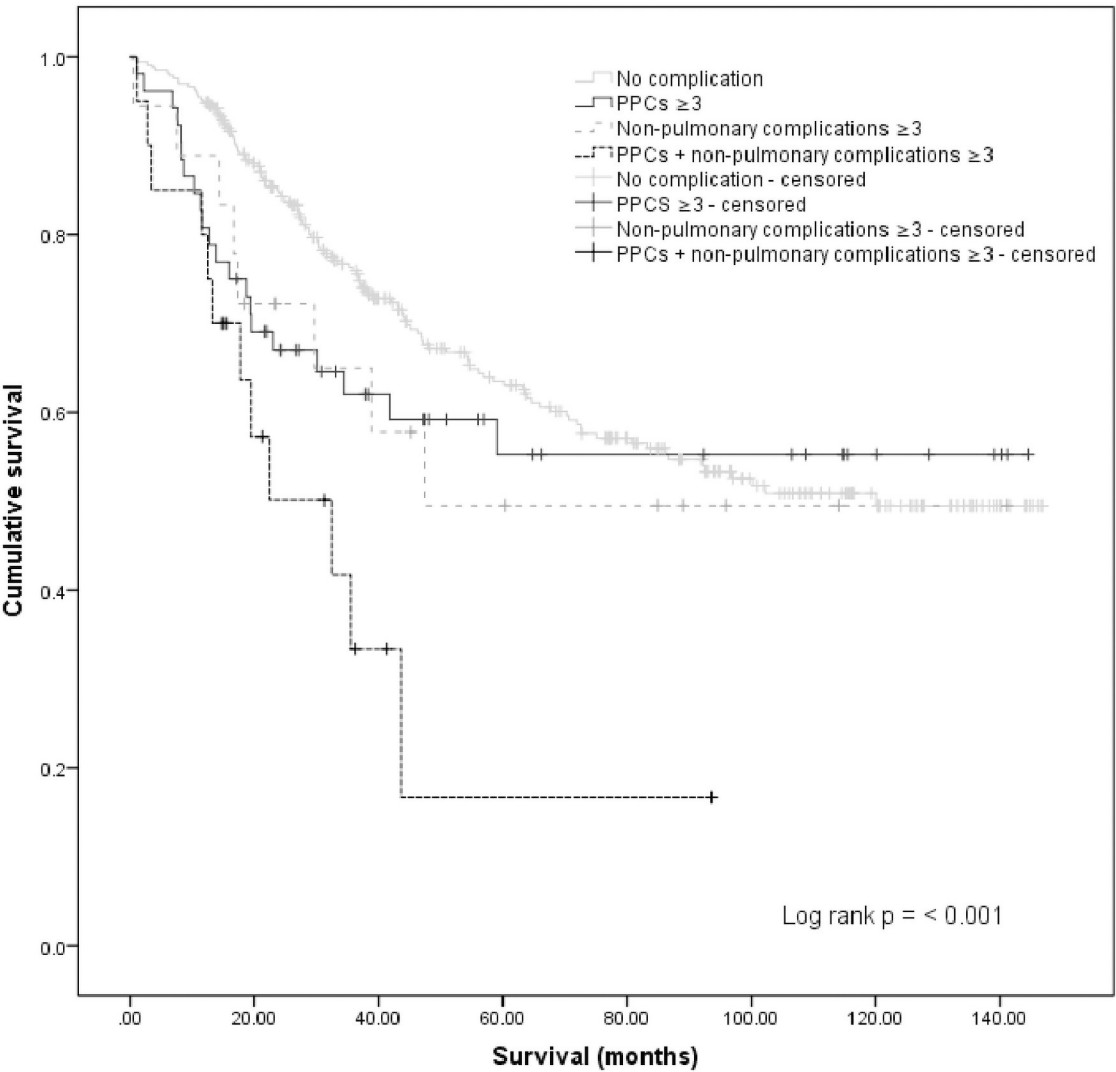

**Fig 2. Kaplan-Meier curves on overall survival in relation to PPCs Grade ≥3, non-pulmonary complications Grade ≥3 or combined PPCs Grade ≥3 and non-pulmonary complications Grade ≥3.** *PPCs* Postoperative pulmonary complications *Non-pulmonary complications* Including abdominal abscess, hemorrhage, cardiac complications, gastrointestinal perforations, sepsis, wound dehiscence, anastomotic leaks and ileus.

system as we did, the Clavien-Dindo classification system, making it difficult to conduct a valid comparison between the results.

The main risk factors in our study were full thickness diaphragmatic injury and/or diaphragmatic resection, both of which are sometimes unavoidable when trying to reach a complete cytoreduction [24–26]. The role of diaphragmatic procedures such as peritonectomy along with full thickness diaphragmatic injury and/or diaphragmatic resection as risk factors for PPCs is controversial [12,13]. In our cohort, diaphragmatic peritonectomy was not a significant risk factor for PPCs, in contrast to previous studies such as the one from Cascales Campos et al. [12] who identified diaphragmatic peritonectomy as the main risk factor. However, they did not find prophylactic chest tubes indicated since the number of patients requiring chest tubes in their study was low. Likewise, Ye et al. [27] did not recommend prophylactic chest tubes after diaphragmatic peritonectomy, but they proposed that they should be

**Table 3. Unadjusted and adjusted results of hazard ratios and p-values for the variables included in the final model on overall survival after CRS and HIPEC.**

| | Unadjusted results | P-value[a] | Adjusted results | P-value[a] |
|---|---|---|---|---|
| | HR (95% CI) | | HR (95% CI) | |
| Gender (male/female) | 0.732 (0.539–0.994) | 0.045 | 0.625 (0.445–0.876) | 0.006 |
| Age | 1.000 (0.999–1.000) | 0.602 | 1.000 (1.000–1.000) | 0.870 |
| BMI | | | | |
| < 18.5 | 2.773 (1.007–7.631) | 0.048 | 4.683 (1.640–13.369) | 0.004 |
| 18.5–25 | Reference | | Reference | |
| > 25–30 | 0.808 (0.572–1.143) | 0.228 | 0.726 (0.503–1.048) | 0.088 |
| > 30 | 1.035 (0.671–1.596) | 0.876 | 0.911 (0.576–1.442) | 0.691 |
| Comorbidity (yes/no) | 0.914 (0.665–1.257) | 0.580 | | |
| ASA score | 1.260 (1.000–1.587) | 0.050 | 1.198(0.931–1.540) | 0.160 |
| Diagnosis (PMP, no neoplastic cells/colorectal, mesothelioma, others) | 7.988 (5.122–12.456) | < 0.001 | 8.085 (5.106–12.801) | < 0.001 |
| CC score (0/1) | 0.775 (0.529–1.136) | 0.192 | 0.581 (0.370–0.913) | 0.019 |
| PCI | 1.021 (1.007–1.035) | 0.003 | 1.035 (1.018–1.053) | < 0.001 |
| Liver resection | 1.060 (0.740–1.520) | 0.750 | | |
| Splenectomy | 0.894 (0.646–1.238) | 0.500 | | |
| No complication | Reference | | Reference | |
| PPCs ≥3 | 1.285 (0.809–2.042) | 0.288 | 1.311 (0.793–2.167) | 0.291 |
| Non-pulmonary complications ≥3 | 1.389 (0.679–2.840) | 0.368 | 1.259 (0.613–2.586) | 0.531 |
| PPCs + non-pulmonary complications ≥3 | 3.328 (1.826–6.064) | < 0.001 | 2.285 (1.232–4.241) | 0.009 |

*CRS* Cytoreductive surgery *HIPEC* Hyperthermic intraperitoneal chemotherapy *HR* Hazard ratio *CI* Confidence interval.

*BMI* Body mass index *ASA score* American Association of Anesthesiologists score *PMP* Pseudomyxoma peritonei.

*CC score* Completeness of cytoreduction score *PCI* Peritoneal cancer index *PPC* Postoperative pulmonary complication.

[a] Cox regression.

considered after full thickness diaphragmatic resection. With the knowledge that severe PPCs, such as pleural effusion requiring chest tubes, might be related to increased mortality, our study suggests a re-evaluation of the use of prophylactic chest tubes. This is supported by Sandadi et al [28] who found that prophylactic chest tubes decreased the incidence of pleural effusion after diaphragmatic procedures as a part of CRS. The management of chest tubes is not without complications since they present the potential for injuries to the stomach and infections [29]. However, tubes inserted intraoperatively under full control are probably safer to use.

Surprisingly, neither smoking nor age, both established risk factors for the development of PPCs [14,30–32], were associated with severe PPCs in this cohort. This is interesting and might be a result of patient selection for CRS and HIPEC where heavy smokers with obstructive pulmonary disease might have been ineligible for surgery. Patients with PM are also younger on average than many other cancer patient groups and this might reduce the impact of age in this cohort. These results concur with previous studies of patients with PM [12,13].

This study adds a new perspective regarding the impact of PPCs on overall survival after CRS and HIPEC. Both increased in-hospital mortality [33] and long-term mortality [34] have been associated with PPCs, but this is the first study to examine the effects in patients with peritoneal metastases. In our cohort, neither PPCs Grade ≥ 3 nor non-pulmonary complications ≥ 3 alone affected survival rates but when combined, they were associated with decreased overall survival. It has not been established whether it is the PPC itself or the effect of multiple complications that contributes to the decreased overall survival and further studies

are required to fully understand the mechanisms behind the increased mortality rates. It would also be useful to investigate whether PPCs are working as a proxy for some other processes in the body affecting mortality.

Some limitations of this study must be considered when interpreting the results. First, data on cause of death were not available in the medical records. Unfortunately, data on recurrence were only partly registered in the medical records since follow up was handled by the referring hospitals. Furthermore, we must also consider that only data on PPCs occurring during the hospital stay at Uppsala University Hospital were accessible. This means that complications occurring later in the postoperative period that might have had an impact on survival rates could have been missed. The incidence rate of severe PPCs presented in this study might also be affected by the short follow-up period. This study suggests that severe PPCs might play a major role for overall survival. However, this needs more thorough investigation.

One of the strengths of our study is that it uses a standardized grading system that is well established in surgical care and all grades of complications are included. This gives a broad picture on the incidence of severe PPCs after CRS and HIPEC and facilitates future comparisons. Another strength of the study is that we included all tumor types, which resulted in a large cohort and a valid representation of the CRS and HIPEC population.

We are unable to draw conclusions on whether it is the PPC itself or the effect of multiple complications that is responsible for the negative correlation with overall survival in this study but regardless of the underlying mechanisms, we suggest that the prevention of severe PPCs can benefit survival rates. It has previously been established that PPCs after abdominal surgery can, to some extent, be prevented [35] and while some complications such as pleural effusion might be rather difficult to avoid when performing diaphragmatic interventions [12,36], others can be more easily prevented. Preoperative physical therapy including inspiratory muscle training (IMT) is one example that has been shown to decrease the risk of pneumonia after major abdominal surgery [35]. There are no studies on physical therapy focused on pre-habilitation performed on patients undergoing CRS and HIPEC, but this might be a promising preoperative intervention to decrease the risk of PPCs and, by extension, it might also affect overall survival.

This study highlights the impact severe PPCs might have on survival rates, but the underlying mechanisms are still unclear. Evaluating the effect of pre-habilitation interventions on PPCs in patients with peritoneal metastases would be an interesting research subject which might shed light on whether the prevention of PPCs, and thus the potential for avoiding decreased overall survival after CRS and HIPEC, is possible.

## Conclusion

Severe PPCs are common after CRS and HIPEC and in our study, full thickness diaphragmatic injury and/or diaphragmatic resection were the most influential risk factors. We also found that having both PPCs Grade ≥ 3 and non-pulmonary complications Grade ≥ 3 was associated with decreased overall survival. These results suggest the need for further research on PPCs and mortality to be able to fully understand the mechanisms.

## Supporting information

**S1 Appendix. Supplementary file regression analyses.**
(DOCX)

## Acknowledgments

The authors thank Patrik Öhagen, Uppsala Clinical Research Center, and Jonas Selling, Academy of Statistics (Statistikakademin), for statistical advice.

## Author Contributions

**Conceptualization:** Olivia Sand, Mikael Andersson, Erebouni Arakelian, Peter Cashin, Egidijus Semenas, Wilhelm Graf.

**Data curation:** Olivia Sand, Mikael Andersson.

**Formal analysis:** Olivia Sand, Mikael Andersson, Peter Cashin, Wilhelm Graf.

**Funding acquisition:** Wilhelm Graf.

**Methodology:** Olivia Sand, Mikael Andersson, Peter Cashin, Egidijus Semenas, Wilhelm Graf.

**Project administration:** Olivia Sand.

**Resources:** Egidijus Semenas, Wilhelm Graf.

**Supervision:** Mikael Andersson, Erebouni Arakelian, Peter Cashin, Egidijus Semenas, Wilhelm Graf.

**Validation:** Olivia Sand, Mikael Andersson, Erebouni Arakelian, Peter Cashin, Egidijus Semenas, Wilhelm Graf.

**Visualization:** Olivia Sand.

**Writing – original draft:** Olivia Sand.

**Writing – review & editing:** Olivia Sand, Mikael Andersson, Erebouni Arakelian, Peter Cashin, Egidijus Semenas, Wilhelm Graf.

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
