## [Editor Report · Decision Letter 0]

26 Jul 2021

PONE-D-21-23694

Severe pulmonary complications after cytoreductive surgery and HIPEC are common and contribute to decreased overall survival

PLOS ONE

Dear Dr. Sand,

Thank you for submitting your manuscript to PLOS ONE. After careful consideration, we feel that it has merit but does not fully meet PLOS ONE’s publication criteria as it currently stands. Therefore, we invite you to submit a revised version of the manuscript that addresses the points raised during the review process.

No uncommon abbreviation or acronyms in the Title, please!

What is "HIPEC"?

Please also do English editing in the main text.

We look forward to receiving your revised manuscript.

Kind regards,

Academic Editor

PLOS ONE
---

## [Author Response · Author response to Decision Letter 0]

24 Sep 2021

1. No uncommon abbreviation or acronyms in the Title, please! What is HIPEC? 

Thank you for pointing this out. We have now made revisions according to the guidelines of PLOS ONE and spelled out the acronym hyperthermic intraperitoneal chemotherapy. We appreciate that you made us aware of this error. 

2. Please also do English editing in the main text.

Thank you for your comment. With the help of a native English reviewer, we have edited the main text to improve the text and we believe that this version of the manuscript is of higher quality. Please find the changes made to the manuscript in “Revised manuscript with tracked changes”. 

We appreciate your comment. In the revised edition of the manuscript please find that we have changed all level 1 headings to bold type with an 18 pt font, all level 2 headings to bold type with a 16 pt font, changed to bold type for all table and figure titles and revised the names of the two figure files and the manuscript file. We have also made sure that the text is double-spaced. 

4. We note that the grant information you provided in the ‘Funding Information’ and ‘Financial Disclosure’ sections do not match. When you resubmit, please ensure that you provide the correct grant numbers for the awards you received for your study in the ‘Funding Information’ section.

Thank you for making us aware of this error. The information provided in the “Funding information” section now matches the information provided in the “Financial Disclosure”. For clarity, we received funding from the Swedish Cancer. Society (Cancerfonden) grant/award number 160411 and Wilhelm Graf was the one who received this research grant.

We very much appreciate your thoroughness in noticing this. The abstract on the online submission has now been revised to completely match the abstract in the manuscript.

6 (added Sept 24, 2021). We note that you have indicated that data from this study are available upon request. Please note PLOS only allows data to be available upon request if there are legal or ethical restrictions on sharing data publicly.

In this instance it seems there may be acceptable restrictions in place, however we ask that you please kindly clarify in detail the reasons for data restriction (e.g., data contain potentially sensitive information, etc.) and who has imposed them (e.g., an ethics committee). Please also provide a non-author point of contact (e.g., data access committee, ethics committee, or other institutional body) where data request may be made. Note that it is not acceptable for the authors to be the sole named individuals responsible for ensuring data access.

Thank you for allowing us to clarify this. We have now added the requested information in the Data availability statement and it reads as follows: The data underlying the results presented in this study contain potentially sensitive and identifying participant information and cannot be shared publicly due to GDPR. The data are available upon request from registrator@uu.se (reference UU-DsO 2021/106).

---

## [Decision Letter · Decision Letter 1]

29 Nov 2021

PONE-D-21-23694R1Severe pulmonary complications after cytoreductive surgery and hyperthermic intraperitoneal chemotherapy are common and contribute to decreased overall survivalPLOS ONE

Dear Dr. Sand,

Thank you for submitting your manuscript to PLOS ONE. After careful consideration, we feel that it has merit but does not fully meet PLOS ONE’s publication criteria as it currently stands. Therefore, we invite you to submit a revised version of the manuscript that addresses the points raised during the review process.

Please revise.

We look forward to receiving your revised manuscript.

Kind regards,

Academic Editor

PLOS ONE

Journal Requirements:

Reviewers' comments:

Reviewer's Responses to Questions

**Comments to the Author**

1. If the authors have adequately addressed your comments raised in a previous round of review and you feel that this manuscript is now acceptable for publication, you may indicate that here to bypass the “Comments to the Author” section, enter your conflict of interest statement in the “Confidential to Editor” section, and submit your "Accept" recommendation.

Reviewer #1: All comments have been addressed

Reviewer #2: All comments have been addressed

2. Is the manuscript technically sound, and do the data support the conclusions?

Reviewer #1: Yes

Reviewer #2: Yes

3. Has the statistical analysis been performed appropriately and rigorously? 

Reviewer #1: Yes

Reviewer #2: I Don't Know

4. Have the authors made all data underlying the findings in their manuscript fully available?

Reviewer #1: Yes

Reviewer #2: No

5. Is the manuscript presented in an intelligible fashion and written in standard English?

Reviewer #1: Yes

Reviewer #2: Yes

6. Review Comments to the Author

Reviewer #1: In the discussion section, line 216 to line 2018, the authors state that "Unfortunately, we could find no previous studies on PPCs in patients with peritoneal metastases using the Clavien-Dindo classification system, thus making it difficult to compare result". I came across another study, using a rather different, but used Clavien-Dindo classification and included postoperative pulmonary complications among other postoperative complications of CRS and HIPEC.

Sinukumar S, Mehta S, Damodaran D, Rajan F, Zaveri S, Ray M, Katdare N, Sethna K, Patel MD, Kammer P, Peedicayil A, Bhatt A. Failure-to-Rescue Following Cytoreductive Surgery with or Without HIPEC is Determined by the Type of Complication-a Retrospective Study by INDEPSO. Indian J Surg Oncol. 2019 Feb;10(Suppl 1):71-79. doi: 10.1007/s13193-019-00877-x. Epub 2019 Jan 14. PMID: 30886497; PMCID: PMC6397122.

I do not think that the findings of that study would add much to the current work in consideration, but I think that, for the sake of perfection, the mentioned sentence should be omitted or the other work get discussed.

Reviewer #2: (No Response)

7. PLOS authors have the option to publish the peer review history of their article (what does this mean?). If published, this will include your full peer review and any attached files.

Reviewer #1: **Yes: **Salah-Eldin Abdelmoneim

Reviewer #2: No

---

## [Author Response · Author response to Decision Letter 1]

7 Dec 2021

The reference list has been reviewed as requested. No article in the reference list has been retracted. 

2. Have the authors made all data underlying the findings in their manuscript fully available?

Reviewer #1: Yes

Reviewer #2: No

We would like to clarify our statement regarding data availability once more. It is due to the General Data Protection Regulation (GDPR) and the potentially sensitive and identifying nature of our data that we cannot share our data publicly. The data are however available upon request from registrator@uu.se (reference UU-DsO 2021/106).

3. Reviewer #1: In the discussion section, line 216 to line 2018, the authors state that "Unfortunately, we could find no previous studies on PPCs in patients with peritoneal metastases using the Clavien-Dindo classification system, thus making it difficult to compare result". I came across another study, using a rather different, but used Clavien-Dindo classification and included postoperative pulmonary complications among other postoperative complications of CRS and HIPEC.

Sinukumar S, Mehta S, Damodaran D, Rajan F, Zaveri S, Ray M, Katdare N, Sethna K, Patel MD, Kammer P, Peedicayil A, Bhatt A. Failure-to-Rescue Following Cytoreductive Surgery with or Without HIPEC is Determined by the Type of Complication-a Retrospective Study by INDEPSO. Indian J Surg Oncol. 2019 Feb;10(Suppl 1):71-79. doi: 10.1007/s13193-019-00877-x. Epub 2019 Jan 14. PMID: 30886497; PMCID: PMC6397122.

I do not think that the findings of that study would add much to the current work in consideration, but I think that, for the sake of perfection, the mentioned sentence should be omitted or the other work get discussed.

Reviewer #2: (No Response)

Thank you for your thorough review and for enlightening us on this study. After reading the article, we have modified the section and the suggested article is mentioned in the revised version. However, in the study by Sinukumar et al., they used the CTCAE classification and not the Clavien-Dindo classification but for the sake of completeness, this article is now discussed and included in the reference list. 

4. (Added Dec 7th) Thank you for including your ethics statement on the online submission form: "The study was approved by the Swedish Ethical Review Authority in Uppsala, Sweden (reference no. 2013/203). The data were analyzed anonymously and therefore written/oral consent was not required or obtained.". To help ensure that the wording of your manuscript is suitable for publication, would you please also add this statement at the beginning of the Methods section of your manuscript file.

Thank you for allowing us to make this revision to our manuscript. We have now added this statement at the beginning of the Methods section both in the main manuscript file and in the manuscript with track changes file. 

We look forward to hearing from you in due time regarding our submission and to respond to any further questions and comments you may have. 

Sincerely, 

Olivia Sand

---

## [Decision Letter · Decision Letter 2]

13 Dec 2021

Severe pulmonary complications after cytoreductive surgery and hyperthermic intraperitoneal chemotherapy are common and contribute to decreased overall survival

PONE-D-21-23694R2

Dear Dr. Sand,

We’re pleased to inform you that your manuscript has been judged scientifically suitable for publication and will be formally accepted for publication once it meets all outstanding technical requirements.

Kind regards,

Academic Editor

PLOS ONE

Additional Editor Comments (optional):

Reviewers' comments:

Reviewer's Responses to Questions

**Comments to the Author**

1. If the authors have adequately addressed your comments raised in a previous round of review and you feel that this manuscript is now acceptable for publication, you may indicate that here to bypass the “Comments to the Author” section, enter your conflict of interest statement in the “Confidential to Editor” section, and submit your "Accept" recommendation.

Reviewer #1: All comments have been addressed

Reviewer #2: All comments have been addressed

2. Is the manuscript technically sound, and do the data support the conclusions?

Reviewer #1: Yes

Reviewer #2: Yes

3. Has the statistical analysis been performed appropriately and rigorously? 

Reviewer #1: Yes

Reviewer #2: Yes

4. Have the authors made all data underlying the findings in their manuscript fully available?

Reviewer #1: Yes

Reviewer #2: Yes

5. Is the manuscript presented in an intelligible fashion and written in standard English?

Reviewer #1: Yes

Reviewer #2: Yes

6. Review Comments to the Author

Reviewer #1: (No Response)

Reviewer #2: (No Response)

7. PLOS authors have the option to publish the peer review history of their article (what does this mean?). If published, this will include your full peer review and any attached files.

Reviewer #1: **Yes: **Salah-Eldin Abdelmoneim

Reviewer #2: No

---

## [Editor Report · Acceptance letter]

17 Dec 2021

PONE-D-21-23694R2 

Severe pulmonary complications after cytoreductive surgery and hyperthermic intraperitoneal chemotherapy are common and contribute to decreased overall survival 

Dear Dr. Sand:

I'm pleased to inform you that your manuscript has been deemed suitable for publication in PLOS ONE. Congratulations! Your manuscript is now with our production department. 

Kind regards, 

on behalf of

Dr. Robert Jeenchen Chen 

Academic Editor

PLOS ONE